# Biomarkers in Adult-Type Diffuse Gliomas: Elevated Levels of Circulating Vesicular Heat Shock Protein 70 Serve as a Biomarker in Grade 4 Glioblastoma and Increase NK Cell Frequencies in Grade 3 Glioma

**DOI:** 10.3390/biomedicines11123235

**Published:** 2023-12-07

**Authors:** Philipp Lennartz, Dennis Thölke, Ali Bashiri Dezfouli, Mathias Pilz, Dominik Lobinger, Verena Messner, Hannah Zanth, Karen Ainslie, Morteza Hasanzadeh Kafshgari, Gerhard Rammes, Markus Ballmann, Martin Schlegel, Gemma Ann Foulds, Alan Graham Pockley, Friederike Schmidt-Graf, Gabriele Multhoff

**Affiliations:** 1Central Institute for Translational Cancer Research Technische Universität München (TranslaTUM), Klinikum rechts der Isar, TUM School of Medicine and Health, 81675 Munich, Germany; philipp.lennartz@tum.de (P.L.); dennis.thoelke@hotmail.de (D.T.); ali.bashiri@tum.de (A.B.D.); ge54may@mytum.de (V.M.); hannah.zanth@tum.de (H.Z.);; 2Department of Radiation Oncology, Klinikum rechts der Isar, TUM School of Medicine and Health, 81675 Munich, Germany; 3Department of Otolaryngology, Head and Neck Surgery, Klinikum rechts der Isar, TUM School of Medicine and Health, 81675 Munich, Germany; 4Department of Thoracic Surgery, München Klinik Bogenhausen, Lehrkrankenhaus der TUM, 81925 Munich, Germany; dominiklobinger@web.de; 5Department of Biomedical Electronics, Central Instititute for Translational Cancer Research, Technische Universität München (TranslaTUM), Klinikum rechts der Isar, TUM School of Medicine and Health, 81675 Munich, Germany; 6Department of Anaesthesiology and Intensive Care Medicine, Klinikum rechts der Isar, TUM School of Medicine and Health, 81675 Munich, Germanymartin.schlegel@tum.de (M.S.); 7John van Geest Cancer Research Centre, School of Science and Technology, Nottingham Trent University, Nottingham NG11 8NS, UK; gemma.foulds@ntu.ac.uk (G.A.F.); graham.pockley@ntu.ac.uk (A.G.P.); 8Department of Neurology, Klinikum rechts der Isar, TUM School of Medicine and Health, 81675 Munich, Germany; f.schmidt-graf@tum.de

**Keywords:** biomarkers, liquid biopsy, extracellular free and vesicular Hsp70, plasma membrane bound Hsp70, glioblastoma, oligodendroglioma, astrocytoma, isocitrate dehydrogenase 1/2, NK cells, immunophenotyping

## Abstract

The presence of circulating Hsp70 levels and their influence on the immunophenotype of circulating lymphocyte subsets were examined as diagnostic/prognostic biomarkers for the overall survival (OS) in patients with IDH-mutant WHO grade 3 oligodendroglioma, astrocytoma, and IDH-wildtype grade 4 glioblastoma (GBM). Vesicular and free Hsp70 in the plasma/serum was measured using the Hsp70-exo and R&D Systems DuoSet^®^ Hsp70 ELISAs. The immunophenotype and membrane Hsp70 status was determined by multiparameter flow cytometry on peripheral blood lymphocytes and single-cell suspensions of tumor specimens and cultured cells. Compared to healthy controls, circulating vesicular Hsp70 levels were significantly increased in patients with GBM, concomitant with a significant decrease in the proportion of CD3+/CD4+ helper T cells, whereas the frequency of NK cells was most prominently increased in patients with grade 3 gliomas. Elevated circulating Hsp70 levels and a higher prevalence of activated CD3−/CD56+/CD94+/CD69+ NK cells were associated with an improved OS in grade 3 gliomas, whereas high Hsp70 levels and low CD3+/CD4+ frequencies were associated with an adverse OS in GBM. It is assumed that a reduced membrane Hsp70 density on grade 4 versus grade 3 primary glioma cells and reduced CD3+/CD4+ T cell counts in GBM might drive an immunosuppressive tumor microenvironment.

## 1. Introduction

Malignant brain tumors comprise approximately 30% of all primary brain tumors, of which 80% are adult-type diffuse gliomas—a group of highly infiltrative and largely incurable malignancies [1]. The 2021 update of the WHO classification of central nervous system tumors (WHO CNS5) identifies tissue biopsy-based molecular biomarkers for defining neoplastic entities. The two most important molecular biomarkers, wildtype or mutant isocitrate dehydrogenase 1/2 (IDH) and the 1p19q codeletion, differentiate adult-type diffuse gliomas into oligodendroglioma (IDH mutant and 1p19q codeleted), astrocytoma (IDH mutant), and GBM (IDH wildtype) [2]. In general, patients with IDH-mutated glioma have a better outcome compared to patients with IDH-wildtype GBM. IDH-mutant gliomas can be further differentiated by the *ATRX*, *CDKN2A/B*, and *TP53* status, whereas the *TERT* mutation, *EGFR* amplification status, *MGMT* promotor methylation status, and whole gain of chromosome 7 and whole loss of chromosome 10 status can further differentiate IDH-wildtype GBM [2,3,4].

Grade 4 GBM is the most common and aggressive primary malignant brain tumor and accounts for approximately 15% of all intracranial neoplasms and 45–50% of all primary malignant brain tumors [1]. The disease course is typically rapid, and despite intensive research and multimodal therapeutic approaches, outcomes remain very poor with a typical median overall survival (OS) of 11–20 months [5,6,7]. It is well known that important factors that contribute to the poor prognosis in GBM are inter- and intratumor molecular and cellular heterogeneity, inadequate drug or agent delivery across the blood brain barrier, redundant and anti-apoptotic signaling pathways, and an immunosuppressive tumor microenvironment (TME) [8,9].

Currently, the diagnosis and classification of diffuse gliomas is mainly based on neuroimaging combined with a histopathological and molecular analysis of tumor biopsies and resected tumor samples [2,10,11]. In addition to tissue biopsies, liquid biopsies have gained importance as a minimally invasive method which is generally well accepted by patients and can be used to identify different molecular biomarkers [12]. With respect to the significant level of heterogeneity in gliomas, especially GBM [8,9], the diagnostic value of tissue samples from singular locations is limited. As a consequence, blood-based biomarkers have particular value for the molecular characterization of diffuse gliomas and the potential to improve patient care in terms of diagnostics, prognostics, and the delivery of adaptive therapies. The major goal of this study was to investigate the potential of molecular and immunological biomarkers in the circulation of glioma patients at initial diagnosis to assess the presence and clinical course of different grade 3 and 4 gliomas. In this regard, levels of the highly conserved, major stress-inducible 70 kD heat shock protein 70 (Hsp70) in the circulation of tumor patients represents a promising candidate. 

It is known that Hsp70 is overexpressed in the cytosol of a large variety of highly aggressive tumor entities, for example, non-small cell lung carcinoma (NSCLC) [13,14], colorectal carcinoma [15,16], and prostate carcinoma [17,18], as well as GBM [19,20,21]. Physiologically, Hsp70 is expressed in nearly all subcellular compartments where it controls protein homeostasis by regulating the maturation and degradation of proteins during cell growth and differentiation, preventing protein aggregation, and transporting other proteins across membranes [22,23]. Cellular stress, such as ischemia, hypoxia, or nutrient deprivation due to increased cell proliferation, which is typical for tumor cells, highly upregulates the synthesis of Hsp70 [23,24]. As cytosolic Hsp70 affects signaling pathways that regulate the apoptosis, proliferation, and differentiation of cells, elevated intracellular Hsp70 levels in tumor cells contribute to stress resistance by preventing apoptotic cell death [22,23,24,25,26].

In addition to its intracellular chaperoning functions, we have previously shown that Hsp70 plays a crucial role as a tumor-specific biomarker when it is presented on the plasma membrane of tumor cells (‘membrane Hsp70’, mHsp70) or when tumor-derived microvesicles expressing mHsp70 are released into the extracellular milieu [20,27,28]. Membrane Hsp70 (mHsp70) expression, which is enabled by a tumor-specific remodeling of the lipid composition [29], stabilizes tumor cell membranes and thereby mediates therapy resistance and increases tumor aggressiveness [30,31] and also serves as a tumor-specific target for activated natural killer (NK) cells [19,32]. With regards to the latter, it has been previously demonstrated that IL-2 and Hsp70-primed CD3−/CD56+ NK cells, which exhibit an increased cell surface density of the C-type lectin receptor complex CD94/NKG2A/C and the activatory receptor CD69, can recognize and kill tumor cells expressing mHsp70 via granzyme B-mediated apoptosis [33]. A randomized phase II clinical trial in patients with advanced NSCLC expressing mHsp70 demonstrated good tolerability and clinical responses after the i.v. injection of autologous NK cells that had been ex vivo activated with low-dose IL-2 and an Hsp70-derived 14mer peptide (‘TKD’) [34]. 

In the circulation of patients with cancer, Hsp70 exists either as a free protein or is expressed on the surface and in the lumen of extracellular lipid microvesicles [20,27,28]. Free Hsp70 predominantly originates from dying or inflamed cells, whereas vesicular Hsp70, which represents the majority of the circulating Hsp70 in patients with cancer, is actively released from viable tumor cells expressing mHsp70 [20,27,35] and has been shown to serve as a biomarker of viable tumor masses in human patients [36]. The newly developed Hsp70-exo enzyme-linked immunoassay (ELISA) can measure both free and vesicular Hsp70 in the peripheral blood of tumor patients [20], whereas the R&D Systems DuoSet^®^ Hsp70 ELISA only detects free Hsp70 [20].

The expression of activating receptors (e.g., C-type lectin receptors and natural cytotoxicity receptors NCRs) enables NK cells to control tumor growth [37], and stress-induced danger-associated molecular patterns (DAMPs) in the circulation can trigger NK cell activity [38,39]. However, in highly aggressive tumor entities, the ability of NK cells to control tumor growth is often impaired by the immunosuppressive nature of the tumor microenvironment (TME) or by antigen loss [40]. The immunosuppressive TME in GBM is considered as one of the key features associated with treatment failure [41,42,43].

Because circulating levels of vesicular Hsp70, as measured using the Hsp70-exo ELISA, are known to correlate with viable tumor masses and to have an impact on the composition of circulating NK cell subpopulations [36,44], the aim of this study was to investigate circulating Hsp70 levels and the immunophenotype of lymphocyte subpopulations, with a focus on NK cells, in the peripheral blood at the initial diagnosis of patients with grade 3 and 4 gliomas. Of particular interest was the identification of a novel chaperone- and immune-related biomarker signature to improve the differentiation of glioma subtypes and to better predict disease prognosis and progression.

## 2. Materials and Methods

### 2.1. Patients

This study was approved by the local ethics committee of the Medical Faculty of the Klinikum rechts der Isar, Technische Universität München (TUM, 324/15s), and written informed consent was obtained from all patients with different types of brain tumors and healthy donors before the start of this study. Blood sampling (EDTA anticoagulation) was conducted in accordance with the Declaration of Helsinki in the Department of Neurology at the Klinikum rechts der Isar, TUM. 

### 2.2. Immunohistochemical Analysis of IDH1 R132H Point Mutation

The *IDH1 R132H* point mutation was determined on deparaffinized and rehydrated formalin-fixed paraffin-embedded brain tumor sections by immunohistochemistry, as recommended by the guidelines. Sections with a high tumor burden were stained with the prediluted IDH1 antibody (DIA-H09-L RTU, Dianova, Hamburg, Germany) using the automated Ventana Benchmark XT stainer. A granular cytosolic staining pattern was considered as IDH-mutant positive. No sequencing was performed in the IDH-mutant-negative cases, but in unclear cases, a positive finding was confirmed by IDH1 and IDH2 sequencing. The *CDKN2A/B* gene status (2021 WHO CNS5) was determined in our hospital from the year 2020 onwards. Since the patients of this study were recruited between 2015 and 2019, the *CDKN2A/B* status was not assessed. 

### 2.3. Measurement of Intracellular Hsp70 by Immunohistochemistry on Tumor Sections

Formalin-fixed paraffin-embedded (FFPE) sections (3 µm) of grade 3 and grade 4 brain tumors were stained with the cmHsp70.1 mAb, as described previously [18]. Briefly, after dewaxing and rehydrating in xylol (Karl Roth GmbH, Karlsruhe, Germany) and descending concentrations of ethanol (96% *v*/*v* Otto Fischar GmbH, Saarbrücken, Germany), sections were heated at 60 °C for 30 min for antigen retrieval. After a washing and blocking step (5% *v*/*v* rabbit serum in antibody diluent, S20022, Agilent Dako, St. Clara, CA, USA), sections were incubated with cmHsp70.1 mAb (1:500, multimmune GmbH, Munich, Germany) at 4 °C overnight. After incubation with an HRP-conjugated secondary antibody and staining with diaminobenzidine (DAB) chromogen, the nucleus was counterstained with hematoxylin and eosin (T865.2, Karl Roth GmbH).

### 2.4. Measurement of Free and Vesicular Hsp70 in the Blood Using the Hsp70-exo ELISA

Plasma prepared from EDTA anticoagulated blood by centrifugation (15 min at 1500× *g*) at 4 °C was aliquoted (300 µL) and stored at −80 °C. Levels of free and vesicular Hsp70 were determined in freshly prepared or frozen plasma samples using the Hsp70-exo ELISA protocol, as described recently [20]. Briefly, 96-well MaxiSorp Nunc Immuno plates (Thermo Fisher, Darmstadt, Germany) were coated overnight with the cmHsp70.2 mAb (1 µg/mL; multimmune GmbH, Munich, Germany) to which, after a washing and blocking step, plasma samples diluted in StabilZyme Select Stabilizer (1:5; Diarect GmbH, Freiburg i. Breisgau, Germany) in duplicate were added alongside an 8-point Hsp70 protein standard (1–100 ng/mL), also diluted in StabilZyme Select (1:5). After washing, plates were incubated with a biotinylated cmHsp70.1 mAb diluted in HRP Protector (200 ng/mL; Candor Biosciences, Candor Biosciences GmbH, Wangen i. Allgäu, Germany) for 30 min. After washing, BioFX-TMB substrate (Surmodics Inc., Eden Prairie, MN, USA) was added and the resulting absorbance read at 450 nm (reference absorbance at 570 nm) using a VICTOR™ X4 Multilabel Plate Reader (PerkinElmer, Waltham, MA, USA).

### 2.5. Measurement of Free Hsp70 in the Blood Using the R&D Systems DuoSet^®^ ELISA

Levels of free Hsp70 in freshly prepared or thawed serum/plasma samples were determined using an R&D Systems DuoSet^®^ Total Hsp70 ELISA (R&D Systems/Bio-Techne GmbH, Wiesbaden-Nordenstadt, Germany) in duplicates, following the manufacturer’s protocol. Colorimetric measurements were performed on a VICTOR™ X4 Multilabel Plate Reader (PerkinElmer, Waltham, MA, USA), as described above. We have previously shown that there are no significant differences in the measured levels of Hsp70 between serum and plasma samples [20].

### 2.6. Exosome Characterization

Extracellular vesicles were isolated from the plasma (10 mL diluted in PBS, 1:1) by sequential centrifugation steps (Optima L-100XP, Beckman Coulter GmbH, Krefeld, Germany) at 300× *g* for 10 min, 2000× *g* for 20 min to remove cellular debris, 12,000× *g* for 45 min to remove other microvesicles, and twice 100,000× *g* for 70 min at room temperature. Freshly isolated extracellular vesicles were resuspended in PBS (pH 7.4) for size and affinity measurements and a flow cytometric analysis. The diameter of the vesicles was determined on a Zetasizer Nano SZ instrument (Malvern, Worcestershire, UK) by dynamic light scattering with a refractive index of 1.38 and an adsorption of 0.01, and the data were analyzed using the FLEX Analysis 3.4 software. The binding affinity (K_D_) of the surface Hsp70-positive extracellular vesicles to cmHsp70.1 mAb (multimmune GmbH, Munich, Germany) was determined by Microscale Thermopheresis (MST) on the Monolith NT.115 (NanoTemper Technologies GmbH, Munich Germany), as described recently [20]. Typical exosomal cell surface markers such as CD9, CD81, and Hsp70 were measured by flow cytometry on a MACSQuant^®^ instrument (Miltenyi Biotec, Bergisch Gladbach, Germany) using fluorescence-labeled mAbs.

### 2.7. Peripheral Blood Immunophenotyping by Multiparameter Flow Cytometry

EDTA anticoagulated blood was immunophenotyped using a BD FACSCalibur™ flow cytometer (BD Biosciences, Heidelberg, Germany). Briefly, 100 µL of EDTA blood was incubated with different panels of fluorescence-labeled antibody combinations. Combining four different fluorescence-labeled antibodies (FITC/PE/PerCP/APC) allowed for the following lymphocyte subpopulations to be determined: CD3−/CD19+ B cells, CD3+/CD45+ T cells, CD3+/CD4+ helper T cells, CD3+/CD8+ cytotoxic T cells, CD3+/CD69+ activated T cells, CD3+/NKG2D+ T cells, CD3+/CD94+ T cells, CD3+/CD4+/CD25+/FoxP3+ regulatory T cells, CD3+/CD8+/CD25+/FoxP3+ regulatory T cells, CD3+/CD56+ NK-like T cells, CD3−/CD56+, CD3−/CD16+, CD3−/CD94+, CD3−/NKG2D+, CD3−/NKp30+, CD3−/NKp46+ NK cells, and CD3−/CD69+ activated NK cells. The antibody combinations used in this study have been described previously [19]. Isotype-matched control antibodies were used as negative controls. The respective percentage of a certain lymphocyte subpopulation is defined as the proportion of cells stained positively for a specific antibody within a defined viable lymphocyte gate.

### 2.8. Flow Cytometry of Membrane-Bound Hsp70 on Viable Primary Brain Tumor Cells, Astrocytes, and Glioblastoma Cell Cultures

The flow cytometric analysis of mHsp70 on viable brain tumor cells and astrocytes has been described previously [19]. In short, tissue biopsies of human brain tumors were digested in a digestion medium (EDTA, 2 mM; L-cysteine 5 mM; papain, 5 U/mL) for 8 min at 37 °C and then forced through a sterile cell strainer (70 µm). Single-cell suspensions of human brain tumor biopsies and cultured primary human astrocytes (InnoProt, Valencia, Spain) and U87MG (ACTT HTB-14) glioblastoma cells were cultured in an astrocyte growth medium (P60101; InnoProt, Valencia, Spain, astrocyte basal medium supplemented with 10% *v*/*v* fetal bovine serum (FBS), astrocyte growth supplement, and penicillin/streptomycin) on poly-L-lysine coated culture flasks. After treatment with trypsin/EDTA for 2 min at 37 °C and a washing step, single cells were incubated with IgG1-FITC/APC/PE (BD Biosciences, Heidelberg, Germany), CD45-APC (FAB114A, R&D Systems, Minneapolis, MN, USA), cmHsp70.1-FITC (multimmune GmbH, Munich, Germany), pan-HLA class I-FITC (F5662, Sigma-Aldrich/Merck, Darmstadt, Germany) mAbs for 30 min at 4 °C in the dark. Cells were incubated with a 7-AAD viability stain (BD Biosciences) before the cytometric analysis to enable the specific analysis of viable cells. Only viable (7-AAD negative) and CD45-negative brain cells were analyzed using a BD FACSCalibur™ flow cytometer (BD Biosciences). The pan-HLA class I antibody staining provides a positive control and the isotype-matched control antibodies a negative control.

### 2.9. Statistical Analysis

Multiple comparisons among all groups were conducted using one-way ANOVA if the data were normally distributed or the Kruskal–Wallis test if the data were not normally distributed. Normal distribution was tested using the Shapiro–Wilk test. The comparison between the percentage of mHsp70-positive tumor cells in grade 3 vs. grade 4 glioma was conducted using the Mann–Whitney test. Comparisons of the Kaplan Meier curves were conducted using the log-rank test. Differences are considered to be statistically significant as follows: not significant, * *p* < 0.05, ** *p* < 0.01, *** *p* < 0.001, **** *p* < 0.0001.

## 3. Results

### 3.1. Patient Characteristics and Overall Survival

Between 2015 and 2019, samples from a total of 130 patients with different brain tumors were included in this clinical study, the aim of which was to identify novel biomarkers for diagnosis and predicting outcomes. As shown in the CONSORT diagram (Figure 1), 31 patients were diagnosed as having WHO grade 3 glioma (*n* = 31), of which 15 were classified as oligodendroglioma (*n* = 15) and 16 as astrocytoma (*n* = 16). Another 99 patients were diagnosed as having WHO grade 4 GBM (*n* = 99). Unexpectedly, none of the grade 4 GBM patients were *IDH R132H* mutant. An explanation for this finding could be the fact that potential cases with non-canonical IDH mutations (i.e., *IDH1 R132S/C/G* or *IDH2 R172* mutations) might have been missed. The male–female ratio of the patient cohort (76 to 23) was higher than the expected ratio of 1.6 to 1. The *CDKN2A/B* status was determined only from the year 2020 onwards in our hospital and, therefore, was not assessed in this patient cohort. Patients with WHO grade 1 or 2 glioma or central nervous system (CNS) metastasis were excluded from this study. EDTA anticoagulated blood was collected from therapy-naïve patients at the initial tumor diagnosis (t = 0) for measuring circulating free Hsp70 levels using the R&D Systems DuoSet^®^ IC Human/Mouse/Rat Total Hsp70 ELISA and free and vesicular Hsp70 levels using the Hsp70-exo ELISA and for the immunophenotyping of major lymphocyte subpopulations. Circulating Hsp70 levels were assessed in the plasma samples of 118 healthy donors (*n* = 118; age range 21–77 years), 10 patients with oligodendroglioma (*n* = 10), 13 patients with astrocytoma grade 3 (*n* = 13), and 76 patients with GBM (*n* = 76). Multiparameter flow cytometric immunophenotyping was conducted in samples from 16 healthy donors (*n* = 16; age range 21–85 years), 13 patients with oligodendroglioma grade 3 (*n* = 13), 15 patients with grade 3 astrocytoma (*n* = 15), and 81 patients with GBM (*n* = 81). Patient characteristics, IDH, 1p19q status, and therapy regimens are summarized in Table 1.

Kaplan–Meier curves of the analyzed patient cohorts show a median overall survival (OS) for patients with oligodendroglioma and astrocytoma of 88 and 32 months, respectively (Figure 2A, *p* = 0.10), a difference which does not reach statistical significance but is in accordance with data derived from other clinical studies [45,46,47]. The median OS of the GBM patients was 11 months, as determined by a Kaplan–Meier analysis, a finding which is also in accordance with other patient cohorts treated with multimodal therapies [5,6,7].

### 3.2. Comparison of Intracellular and Circulating Hsp70 Levels in Patients with Oligodendroglioma, Astrocytoma, and Glioblastoma (GBM) and Characterization of Extracellular Vesicles

Representative views of intracellular Hsp70 levels in tumor areas of FFPE sections (3 µm) derived from grade 3 and grade 4 gliomas determined by immunohistochemical staining with the cmHsp70.1 mAb revealed a higher Hsp70 staining intensity in grade 4 GBM (Figure 3). The surrounding normal brain tissue in grade 4 GBM showed a very low Hsp70 staining intensity. In line with previous data [18], grade 3 gliomas show predominantly a nuclear staining pattern, whereas grade 4 GBM shows a positive staining in both the nucleus and cytosol.

The concentration of free and vesicular Hsp70 in the plasma, as measured using the Hsp70-exo ELISA, is significantly elevated in patients with GBM (**** *p* < 0.0001) compared to the healthy donors (median Hsp70 levels: 45.4 ng/mL (GBM) versus 18.9 ng/mL (healthy) (Figure 4A). The median circulating Hsp70 levels in patients with astrocytoma (27.5 ng/mL) and oligodendroglioma (14.8 ng/mL) were not significantly different to those in the healthy individuals.

The concentration of free Hsp70 in the serum samples, as measured with the R&D Systems DuoSet^®^ Total Hsp70 ELISA, was also significantly elevated in patients with GBM (** *p* < 0.01) compared to the healthy individuals (Figure 4B), with median Hsp70 levels of 3.5 ng/mL (GBM) and 2.6 ng/mL (healthy), respectively. The median levels of free Hsp70 in patients with oligodendroglioma and astrocytoma were 2.4 ng/mL and 3.4 ng/mL, respectively, and these levels were not significantly different to those in the healthy control cohort. Overall, the level of circulating vesicular Hsp70 (Figure 4A) in tumor patients is significantly higher than the level of free Hsp70 (Figure 4B).

The characteristics of the extracellular vesicles derived from a patient with brain tumor were determined by measuring the size by dynamic light scattering using the Zeta sizer (Figure 4C), the binding affinity (Figure 4D) of Hsp70 carrying exosomes to cmHsp70.1 mAb, and typical exosomal surface markers (Figure 4E) such as the tetraspanins CD9, CD81, and mHsp70 by flow cytometry. A representative example of the average diameter (100 nm), K_D_ (9.3 nM), and the surface expression of CD9 (27.3%), CD81 (29.4%), and Hsp70 (98.2%) are illustrated in Figure 4C–E. The Hsp70 value determined with the Hsp70-exo ELISA in the exosomal fraction of this patient was 129.5 ng/mL and that of the exosome-depleted fraction of the same patient was 14.8 ng/mL. 

### 3.3. Immunophenotyping of Major Lymphocyte Subpopulations in the Peripheral Blood of Patients with Oligodendroglioma, Astrocytoma, and Glioblastoma (GBM)

A multiparameter flow cytometric analysis of whole blood samples from 109 patients with glioma using the fluorescence-labeled mAbs described in the Materials and Methods section was undertaken to measure the frequencies of the following major lymphocyte subpopulations: CD3−/CD19+ B cells, CD3+/CD45+ T cells, CD3+/CD4+ helper T cells, CD3+/CD8+ cytotoxic T cells, CD3+/CD69+ activated T cells, CD3+/NKG2D+ T cells, CD3+/CD94+ T cells, CD3+/CD4+/CD25+/FoxP3+ regulatory T cells, CD3+/CD8+/CD25+/FoxP3+ regulatory T cells, CD3+/CD56+ NK-like T cells, CD3−/CD56+, CD3−/CD16+, CD3−/CD94+, CD3−/NKG2D+, CD3−/NKp30+, CD3−/NKp46+ NK cells, and CD3−/CD69+ activated NK cells. Differences in the composition of the lymphocyte subpopulations were seen only in CD3+/CD4+ helper T cells, CD3+/CD8+ cytotoxic T cells, CD3−/CD56+ NK cells, CD56+/CD94+ NK cells, and CD56+/CD69+ NK cells when compared to the healthy control cohort (Figure 5). 

The frequency of CD3+/CD4+ helper T cells in the peripheral blood of all patients with different malignant brain tumor entities was lower than that in the healthy control cohort (Figure 5A). Patients with GBM exhibited a significantly lower frequency of CD3+/CD4+ T cells than that in the healthy control (median values of 39.5% (GBM) versus 47.9% (healthy donors), ** *p* < 0.001). Although patients with oligodendroglioma (median 35.0%; *p* = 0.066) and astrocytoma (median 39.5%; *p* = 0.085) also exhibited reduced proportions of CD3+/CD4+ helper T cells compared to the healthy control cohort, these differences did not reach statistical significance.

Although the median frequency of CD3+/CD8+ cytotoxic T cells (Figure 5B) was marginally increased in the blood of patients with oligodendroglioma, astrocytoma, and GBM (median frequencies of 25.2%, 25.1%, and 19.4%, respectively), no significant differences were observed when compared to the healthy controls (median frequency of 18.4%).

The prevalence of CD3−/CD56+ NK cells (Figure 5C) is similar in patients with oligodendroglioma, astrocytoma, and GBM (median frequencies of 7.9%, 10.7%, and 7.7%, respectively), with no significant differences to the frequency of this cell type in the healthy controls (median frequency of 11.1%). In contrast, the frequency of activated CD56+/CD69+ NK cells is significantly increased in patients with astrocytoma (** *p* < 0.01) and GBM (** *p* < 0.01) compared to the healthy controls (median frequencies of 3.1% (astrocytoma) and 2.5% (GBM) versus 0.9% (healthy donors)). Patients with oligodendroglioma also showed a slightly elevated frequency of CD56+/CD69+ NK cells (median 2.3%); however, this difference did not reach statistical significance.

Patients with oligodendroglioma, astrocytoma, and GBM exhibited elevated median frequencies of CD56+/CD94+ NK cells compared to the healthy controls (6.9%, 7.4%, and 5.6%, respectively, versus 4.5% in the healthy control cohort, Figure 5D). Although this trend was similar to that observed for CD56+/CD69+ NK cells, no statistically significant differences were reached.

### 3.4. Correlation of Hsp70^low^/Hsp70^high^ Levels and Frequencies of NK Cell Subpopulations in Patients with Grade 3 and Grade 4 Gliomas

To investigate the potential relationship between and influence of circulating Hsp70 levels on the composition of NK cell subpopulations in the peripheral blood, we segregated patients with grade 3 and grade 4 glioma into groups with low (Hsp70^low^) and high (Hsp70^high^) circulating Hsp70 levels according to the median circulating Hsp70 levels of each cohort. Patients with grade 3 glioma with Hsp70 plasma levels below 29.8 ng/mL were classified as Hsp70^low^, and patients above this value were classified as Hsp70^high^. For the grade 3 glioma patients, the Hsp70^low^ group comprises 4 patients with oligodendroglioma and 7 patients with astrocytoma (*n* = 11), whereas the Hsp70^high^ group comprises 5 patients with oligodendroglioma and 6 patients with astrocytoma (*n* = 11). Patients with grade 4 glioma, which were all diagnosed as GBM, were considered as Hsp70^low^ (*n* = 30) when their plasma Hsp70 levels were below 45.4 ng/mL (Figure 4A), and patients with plasma levels above this value were considered as Hsp70^high^ (*n* = 32). The Hsp70^low^ and Hsp70^high^ groups were subsequently compared with respect to the frequency of CD3+/CD4+ helper T cells, CD3+/CD8+ cytotoxic T cells, CD3−/CD56+, CD56+/CD94+, and CD56+/CD69+ NK cells. Significant differences in the lymphocyte frequencies between the Hsp70^low^ and Hsp70^high^ glioma patients were observed for CD3−/CD56+, CD56+/CD94+, and CD56+/CD69+ NK cells (Figure 6). None of the other studied major lymphocyte populations showed significant differences between the Hsp70^low^ and Hsp70^high^ groups of patients.

Patients with Hsp70^high^ grade 3 glioma exhibited a significantly elevated frequency of CD3−/CD56+ NK cells (Figure 6A) compared to the Hsp70^low^ patient group (* *p* < 0.05), with median CD3−/CD56+ NK cell frequencies of 15.1% versus 6.9%, respectively. The frequency of CD3−/CD56+ NK cells in patients with Hsp70^high^ and Hsp70^low^ grade 4 GBM did not show any significant differences (Figure 6A).

There were no statistically significant differences in the frequency of CD56+/CD94+ NK cells in the peripheral blood of patients with Hsp70^low^ and Hsp70^high^ grade 3 glioma (Figure 6B). Nevertheless, the frequency of CD56+/CD94+ NK cells in patients with Hsp70^high^ grade 3 glioma were significantly higher than those in the healthy controls (* *p* < 0.05), whereas there was no significant difference between patients with Hsp70^low^ grade 3 glioma and the controls. Similarly, the frequency of CD56+/CD69+ NK cells in patients with Hsp70^high^ grade 3 glioma was significantly higher than that in the controls (** *p* < 0.01, Figure 6B), whereas there was no difference between patients with Hsp70^low^ grade 3 glioma and the controls. Hsp70^high^ and Hsp70^low^ patients with grade 4 GBM did not exhibit any significant differences in the frequencies of CD56+/CD94+ or CD56+/CD69+ NK cells (Figure 6A–C).

A comparison of the frequencies of the NK cell subpopulations indicated above in patients with oligodendroglioma and astrocytoma with low and high circulating Hsp70 levels revealed significantly higher frequencies in CD3−/CD56+, CD56+/CD94+, and CD56+/CD69+ NK cells in patients with oligodendroglioma and high Hsp70 levels compared to those with low Hsp70 levels and/or with healthy individuals (Appendix A).

### 3.5. Correlation of Hsp70^low^/Hsp70^high^ Levels, Lymphocyte Frequencies, and Overall Survival in Patients with Oligodendroglioma, Astrocytoma, and Glioblastoma (GBM)

In addition to its potential use as a diagnostic marker for glioma (Figure 4), circulating Hsp70 levels may also provide a prognostic value, particularly for patients with GBM who exhibit significantly elevated levels of circulating Hsp70 (Figure 4) compared to healthy individuals. To investigate the association of circulating Hsp70 levels on the Overall Survival (OS) of patients with GBM, we segregated the GBM patients into groups exhibiting low and high levels of circulating Hsp70 (Hsp70^low^ and Hsp70^high^) according to the median circulating Hsp70 levels in each cohort. GBM patients with circulating Hsp70 levels below a concentration of 45.4 ng/mL were considered Hsp70^low^ (*n* = 32), whereas patients above this concentration were considered as being Hsp70^high^ (*n* = 27). Although not statistically different, the Hsp70^low^ patients had a median survival of 11 months, whereas the Hsp70^high^ patients had a median survival of 10 months (Figure 7, *p* > 0.05). Patients with grade 3 glioma, who generally had lower circulating Hsp70 levels than the patients with GBM, did not show any differences in the OS when classified into Hsp70^low^ and Hsp70^high^ groups.

To determine possible associations between the frequencies of certain lymphocyte subpopulations and the OS of patients with glioma, patients with grade 3 and grade 4 glioma were segregated into groups having low or high frequencies of different lymphocyte subpopulations such as CD3+/CD4+ helper T cells, CD3+/CD8+ cytotoxic T cells, CD3−/CD56+, CD56+/CD94+, and CD56+/CD69+ NK cells. The cut-off value for the low and high groups corresponds to the median lymphocyte frequency of each group. In the grade 3 glioma patients, significant differences in the OS were only observed for low and high CD3−/CD56+ and CD56+/CD94+ NK cells (Figure 8), but not for other lymphocyte subpopulations.

Patients with grade 3 glioma having a frequency of CD3−/CD56+ NK cells below a cut-off value of 10% were considered as part of the low CD3−/CD56+ NK cell group (*n* = 14; comprising 8 oligodendroglioma and 6 astrocytoma patients), whereas patients with a frequency of these NK cells above 10% were considered as being in the high CD3−/CD56+ NK cell group (*n* = 14; 5 oligodendroglioma, 9 astrocytoma). Patients with grade 3 glioma exhibiting a low frequency of CD3−/CD56+ NK cells had a median OS of 60 months, whereas patients with a high frequency of CD3−/CD56+ NK cells had a median OS of 88 months (Figure 8A, *p* > 0.05). Due to the relatively small patient cohorts, the differences in the OS did not reach statistical significance. In contrast, the patients with grade 4 GBM showed no differences in OS, irrespective of the frequency of CD3−/CD56+ NK cells in the peripheral blood.

Similar results were observed for CD56+/CD94+ NK cells. Patients with grade 3 glioma exhibiting a frequency of CD56+/CD94+ NK cells below a cut-off value of 7.2% were considered as the low CD56+/CD94+ NK cell group (*n* = 14; 7 oligodendroglioma, 7 astrocytoma), whereas patients with a frequency of CD56+/CD94+ NK cells above 7.2% were considered as the high CD56+/CD94+ NK cell group (*n* = 14; 6 oligodendroglioma, 8 astrocytoma). Patients with a low frequency of CD56+/CD94+ NK cells had a median OS of 50 months, whereas patients with a high frequency of CD56+/CD94+ NK cells had a median OS of 88 months (Figure 8B, *p* > 0.05). Patients with grade 4 GBM showed no differences in OS, irrespective of their CD56+/CD94+ NK cell frequency.

### 3.6. Differences in the Expression of Membrane-Bound Hsp70 in Grade 3 and Grade 4 Gliomas, Astrocytes, and Cultured Glioblastoma Cells

The percentage of viable tumor cells expressing mHsp70, as measured by flow cytometry using the cmHsp70.1 mAb, is significantly higher in grade 3 gliomas (94.3%) compared to grade 4 GBM (64.7%, ** *p* < 0.01) (Figure 9). All grade 3 gliomas included show an IDH mutation, whereas all grade 4 GBM are IDH-wildtype. A comparison of the mHsp70 expression on grade 3 oligodendroglioma and astrocytoma revealed no statistically significant differences (Appendix A).

However, a comparison of the mHsp70 expression as determined by flow cytometry on cultured U87MG glioblastoma cells and primary human astrocytes revealed an mHsp70 positivity on viable U87MG cells (61.5%) but not on primary astrocytes (1.7%) (Appendix A). Moreover, circulating Hsp70 was detectable only in the supernatant of cultured U87MG cells but not in that of primary human astrocytes, as determined by ELISA. These data indicate that soluble Hsp70 originates from membrane Hsp70-positive glioma cells but not from primary human astrocytes.

## 4. Discussion

Despite multimodal therapeutic approaches such as maximal resection and adjuvant radiochemotherapy, the mortality of intracranial gliomas remains disproportionately high [48]. The lack of well-established biomarkers for diffuse adult-type gliomas highlights the need for a tumor-specific, universally applicable biomarker which can be measured in liquid biopsies. The expression of the major stress-inducible heat shock protein 70 (Hsp70) is uniquely elevated in a variety of tumor types, including GBM [19,20,21]. Al-though damaged or necrotic tumor cells release free Hsp70 from subcellular compartments into the surrounding tumor environment, a significant amount of Hsp70 is actively released by viable tumor cells in the form of vesicular Hsp70 [20,27,28].

The Hsp70-exo ELISA used in this study utilizes the unique ability of the cmHsp70.1 and cmHsp70.2 mAbs to detect circulating levels of both free and vesicular Hsp70 derived from mHsp70-positive tumor cells in the form of a liquid biopsy [20]. We show that patients with GBM exhibit significantly higher levels of free and vesicular circulating Hsp70 when compared to a healthy cohort, a result which is in line with previous studies [19,20,21]. Crucially, we show that vesicular Hsp70 represents the vast majority of the total circulating Hsp70 in all patient groups studied, including oligodendroglioma, astrocytoma, and GBM, thereby reinforcing the potential value of biomarkers expressed on the surface of extracellular vesicles such as exosomes for gliomas. This finding suggests that the blood–brain barrier does not pose a large obstacle to the measurement of vesicular Hsp70 in liquid biopsies for gliomas that are localized within the central nervous system. As free Hsp70 can be elevated in a variety of systemic inflammatory diseases such as COPD or liver cirrhosis [14,49], membrane-bound Hsp70, selectively expressed on the surface of tumor cells and extracellular vesicles, may be key in differentiating between inflammation and malignant cancer diseases.

As known from other studies, elevated levels of circulating Hsp70 are not only indicative of the presence of a tumor but also correlate with the gross tumor volume, aggressiveness, and cancer stage [20,36]. For tumors of the central nervous system, the extent of damage to the blood–brain barrier may be another, tumor-specific event which results in elevated levels of circulating Hsp70 in different glioma types and stages of disease [50]. In line with this, we show that the elevation of Hsp70 is greater in grade 4 GBM when compared to grade 3 glioma, and this might be reflective of the more aggressive nature of this tumor grade. No differences were observed in circulating Hsp70 levels in grade 3 oligodendroglioma and astrocytoma. Additionally, patients with grade 4 GBM and elevated levels of circulating Hsp70 at the time of initial tumor diagnosis show a trend towards a lower overall survival, compared to patients with initially low Hsp70 levels. This reinforces the proposition that Hsp70 levels in patients with grade 4 GBM might also have prognostic implications, which has previously been shown for patients with colon carcinoma [51,52]. In this regard, a measurement of Hsp70 in regular intervals after tumor diagnosis, which is possible when using a liquid biopsy-based approach, could provide more insight into the development of the disease over time and might offer interesting and potentially important prospects for earlier therapy adaptations.

The overexpression of Hsp70 imparts several, in part conflicting, functions to tumor cells [28]. As a molecular chaperone, Hsp70 is integral to the protein folding, correction, or removal of misfolded proteins and the prevention of protein aggregation, all of which are vital functions for tumor cells experiencing high levels of cellular stress [22,23,24]. Additionally, cytosolic Hsp70 is involved in the inhibition of apoptotic pathways in aberrant tumor cells [22,23,24,25,26]. Membrane-bound Hsp70 stabilizes membranes and mediates therapy resistance [30,31]. On the other hand, membrane-bound Hsp70 contributes to the activation of a patient’s immune system and thereby tumor control by stimulating lymphocytes via antigen cross-presentation [44,53]. In contrast to T cells, NK cells play a pivotal role in the ability of a patient’s immune system to provide early tumor defense without prior antigen stimulation [54]. The targeted response of NK cells to membrane-bound Hsp70 on viable tumor cells has been shown previously [32,33,34]. One identified pathway is the interaction between mHsp70 and the heterodimer of the CD94 lectin receptor and NKG2A or NKG2C on NK cells primed with the TKD 14mer peptide (TKDNNLLGRFELSG, aa 450–463), a sequence derived from the Hsp70 protein expressed on the membrane of tumor cells [32,33,34]. Ultimately, Hsp70-primed NK cells in the presence of IL-2 induce cell death via granzyme B-mediated apoptosis [33,40]; however, associations with NKG2A, NKG2C, NKG2D, and the natural cytotoxicity receptors (NCRs) NKp30, NKp44, and NKp46 [55] suggest that a variety of both activatory and inhibitory pathways can be initiated via membrane-bound Hsp70. The response ultimately depends on the immunocompetence of each individual patient and the TME, as an adequate proinflammatory environment, including factors such as the interleukins IL-2 and IL-15, has been shown to amplify the stimulatory effect of membrane-bound Hsp70 on NK cells [56,57,58]. Interleukin 1 is upregulated in GBM and exhibits suppressing effects mediated by IL-2 [59,60,61,62]. Other factors such as IL-10 and TGF-β and cell surface changes such as the upregulation of PD-L1 may further inhibit tumor control by NK cells in GBM [63,64,65]. A significant reduction in CD4+ helper T cells, as shown in this study, contributes to the immunosuppressive effects induced by GBM.

Although both, grade 3 glioma and grade 4 GBM, show membrane positivity for Hsp70 [19], the flow cytometry of tumor cell suspensions using the cmHsp70.1 mAb allows us to quantify the percentage of tumor cells expressing Hsp70 on the cell membrane. We show that the percentage of tumor cells expressing mHsp70 is greater in grade 3 glioma compared to grade 4 GBM. This finding is in line with data showing that mutations of the tumor suppressor protein p53, which are more frequently found in grade 4 GBM compared to grade 3 glioma, are associated with a reduced Hsp70 expression [66,67]. Antigen presenting MHC class I and II molecules can be downregulated in invasive GBM to escape the T cell-mediated immune responses [68]. We show that mHsp70 follows this trend and may induce similar immunosuppressive effects towards NK cells together with decreased CD3+/CD4+ T cell counts.

Patients with grade 3 glioma retain a greater ability to stimulate NK cells, possibly due to a higher mHsp70 expression density on their tumor cells in a less immunosuppressive TME. Patients with grade 3 glioma expressing high levels of Hsp70 exhibit a significantly increased frequency of CD3−/CD56+ NK cells, thereby supporting the potency of mHsp70 and vesicular Hsp70 in stimulating NK cells in these patients. Additionally, an improved OS in grade 3 glioma patients was found to be associated with a higher frequency of CD3−/CD56+/CD94+ NK cells and an upregulated expression of the activatory marker CD69 on NK cells. These data suggest that activated NK cells can effectively target mHsp70-positive tumor cells and thereby slow the disease progression. In patients with grade 4 GBM with significantly decreased proportions of CD4+ T cells and a highly immunosuppressed TME, elevated Hsp70 levels in the circulation are unable to stimulate an Hsp70-mediated NK cell activity.

The heterogeneous group of brain tumors exhibit very different outcomes in terms of survival and therapeutic response that can be estimated by WHO grading and molecular biomarkers [2,3,4]. An increased expression of mHsp70 on the cell surface of tumor cells may be beneficial for the recognition by activated NK cells to control tumor growth in grade 3 oligodendroglioma and astrocytoma and thereby increasing OS, whereas lower levels of mHsp70 expression and an immunosuppressive TME [9,69,70,71] may inhibit this effect in patients with grade 4 GBM. In a randomized phase II clinical trial, we have shown previously that the adoptive transfer of ex vivo Hsp70-peptide TKD and IL-2-activated autologous (patient-derived) NK cells provides a survival benefit in patients with advanced non-small-cell lung carcinoma [34]. Since these data have been confirmed in a rat model of GBM [57], we assume that Hsp70-activated NK cells may also provide similarly beneficial effects in human glioma patients.

## 5. Conclusions

In summary, we show that Hsp70 in the circulation and on the membrane of tumor cells, in the presence of IL-2, stimulates NK cells in patients with grade 3 glioma and possibly confers a survival benefit to these patients, whereas in patients with grade 4 GBM with low CD4+ helper T cells, the NK cell stimulation is hindered. However, in the latter patients, circulating levels of Hsp70 are significantly elevated from the onset of the tumor disease and may serve as a diagnostic and prognostic tumor marker.

## 6. Patents

The Hsp70-exo ELISA is patented by multimmune GmbH, Munich, Germany (US 11,460,472—other applications pending).

## Figures and Tables

**Figure 1 biomedicines-11-03235-f001:**
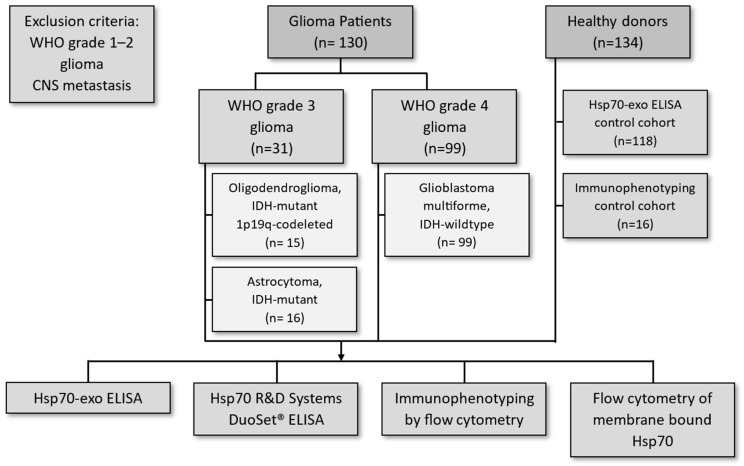
CONSORT diagram of the brain tumor patients and healthy controls included in this study.

**Figure 2 biomedicines-11-03235-f002:**
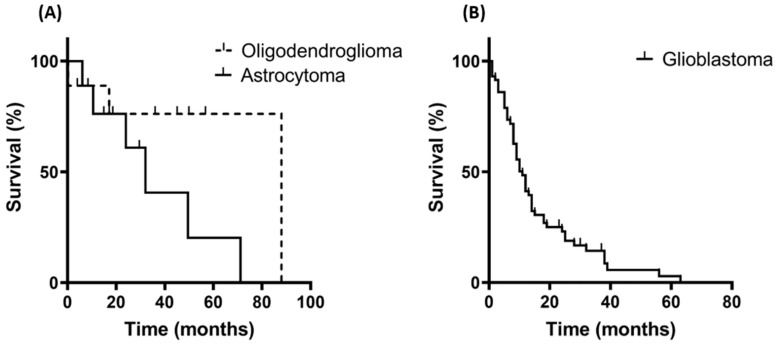
Kaplan–Meier curves of the overall survival (OS) of patients with grade 3 and grade 4 glioma. (**A**) OS of patients with oligodendroglioma (*n* = 15; median survival 88 months) and astrocytoma (*n* = 16; median survival 32 months). (**B**) OS of patients with glioblastoma (GBM; *n* = 78; median survival 11 months).

**Figure 3 biomedicines-11-03235-f003:**
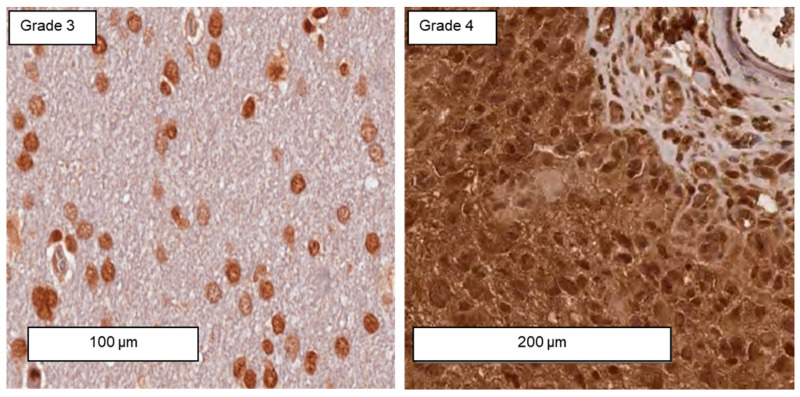
Representative immunohistochemical staining of intracellular Hsp70 levels in tumor sections of grade 3 and grade 4 gliomas using cmHsp70.1 mAb. Scale bars, 100 µm and 200 µm.

**Figure 4 biomedicines-11-03235-f004:**
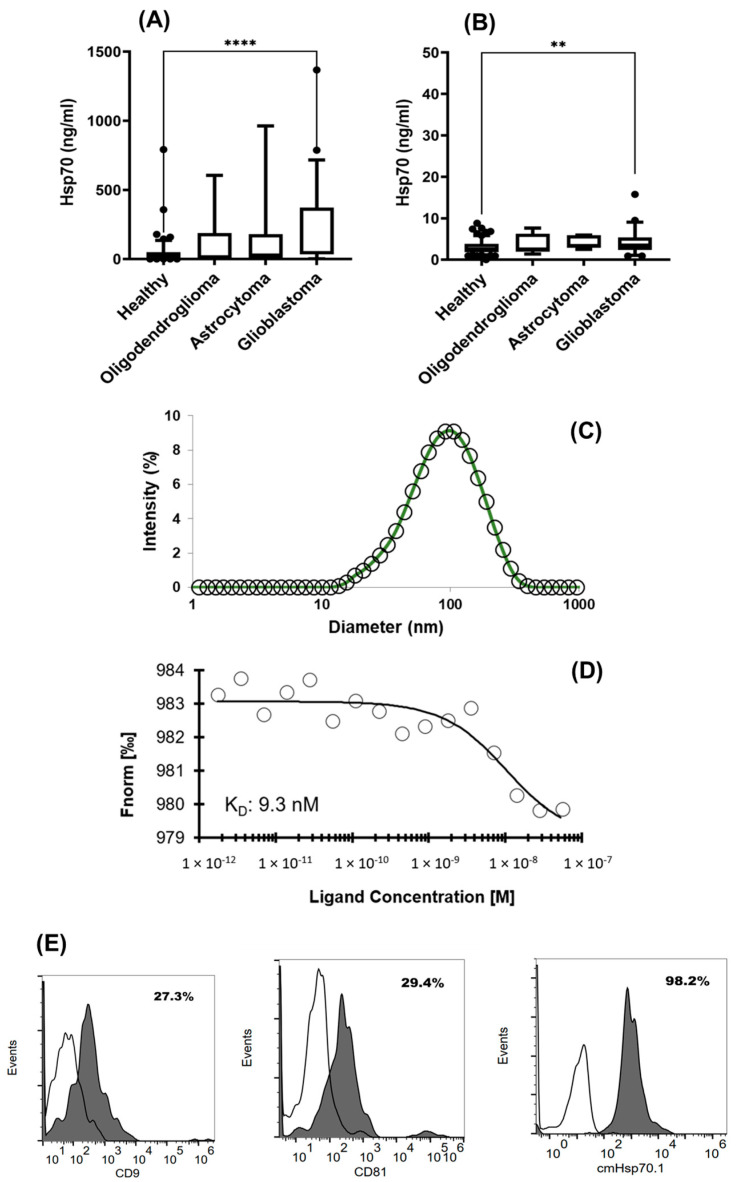
(**A**) Free and vesicular Hsp70 (ng/mL) levels measured in the plasma of healthy donors (*n* = 118; Hsp70 median, 18.9 ng/mL) and patients with oligodendroglioma (*n* = 10; Hsp70 median, 14.8 ng/mL), astrocytoma (*n* = 13; Hsp70 median, 27.5 ng/mL), and GBM (*n* = 76; Hsp70 median, 45.4 ng/mL), as measured using the Hsp70-exo ELISA. (**B**) Free Hsp70 levels measured in the serum of healthy donors (*n* = 150; Hsp70 median, 2.6 ng/mL) and patients with oligodendroglioma (*n* = 7; Hsp70 median, 2.4 ng/mL), astrocytoma (*n* = 5, Hsp70 median, 3.4 ng/mL), and GBM (*n* = 51; Hsp70 median, 3.5 ng/mL) using the DuoSet^®^ IC Human/Mouse/Rat Total Hsp70 ELISA. Statistically significant differences ** *p* < 0.01, **** *p* < 0.0001. (**C**) Size distribution of extracellular vesicles, (**D**) binding affinity (K_D_) of extracellular vesicles to cmHsp70.1 mAb, and (**E**) typical exosomal surface markers such as CD9, CD81, and Hsp70 on extracellular vesicles derived from a patient with astrocytoma.

**Figure 5 biomedicines-11-03235-f005:**
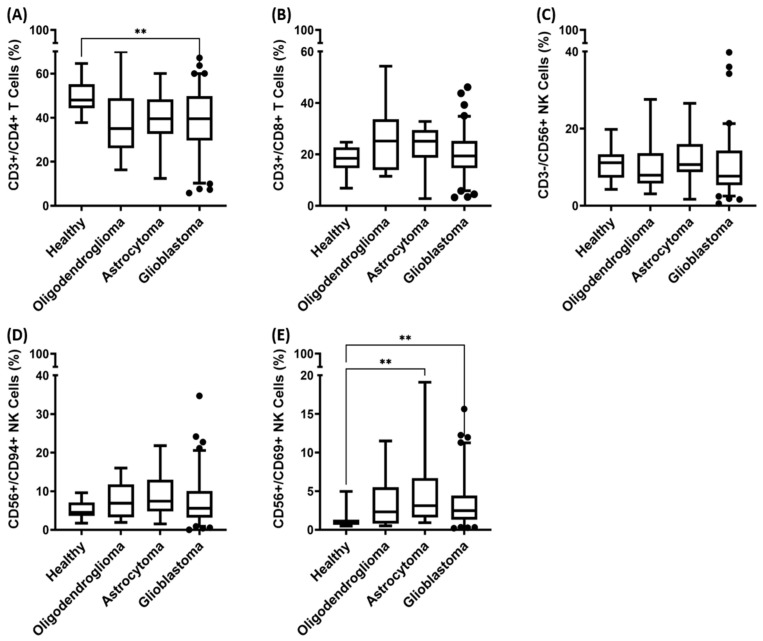
Proportions of lymphocyte subsets in the peripheral blood of healthy donors (*n* = 16) and patients with oligodendroglioma (*n* = 13), astrocytoma (*n* = 15), and glioblastoma (GBM) (*n* = 81). (**A**) CD3+/CD4+ helper T cells, as determined by multiparameter flow cytometry. (**B**) CD3+/CD8+ cytotoxic T cells, (**C**) CD3−/CD56+ NK cells, (**D**) CD56+/CD94+ NK cells, and (**E**) CD56+/CD69+ NK cells. Statistically significant differences ** *p* < 0.01.

**Figure 6 biomedicines-11-03235-f006:**
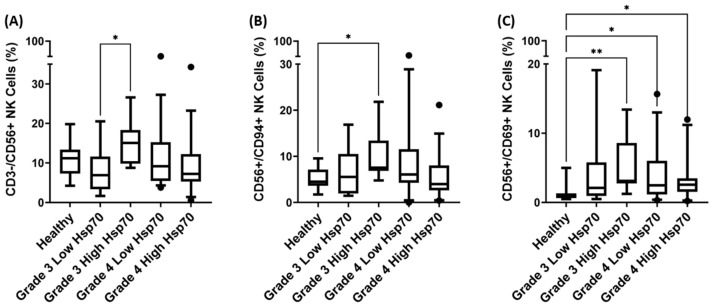
Correlation of low and high Hsp70 values in the circulation and the proportion of different NK cell subpopulations in patients with grade 3 glioma (Hsp70^low^ *n* = 11; Hsp70^high^ *n* = 11) and grade 4 GBM (Hsp70^low^ *n* = 30; Hsp70^high^ *n* = 32) compared to a healthy control cohort (*n* = 16). (**A**) CD3−/CD56+ NK cells in grade 3 and grade 4 GBM patients with high (Hsp70^high^) and low (Hsp70^low^) Hsp70 plasma levels. (**B**) CD56+/CD94+ NK cells in grade 3 and grade 4 GBM patients with high (Hsp70^high^) and low (Hsp70^low^) Hsp70 plasma levels. (**C**) CD56+/CD69+ NK cells in grade 3 and grade 4 GBM patients with high (Hsp70^high^) and low (Hsp70^low^) Hsp70 plasma levels. Statistically significant differences * *p* < 0.05, ** *p* < 0.01.

**Figure 7 biomedicines-11-03235-f007:**
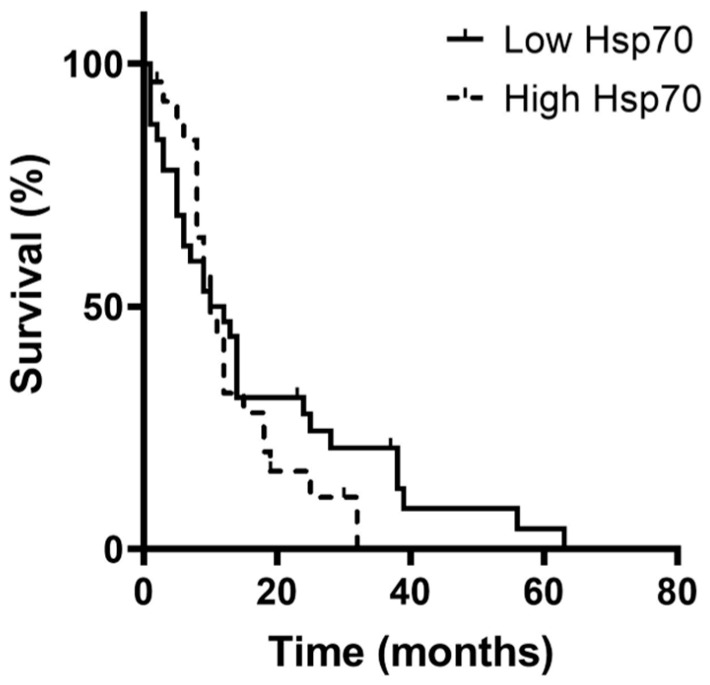
Correlation of circulating Hsp70^low^ and Hsp70^high^ values and overall survival (OS) in patients with grade 4 GBM (low, *n* = 32; high, *n* = 27), as determined by the Kaplan–Meier analysis.

**Figure 8 biomedicines-11-03235-f008:**
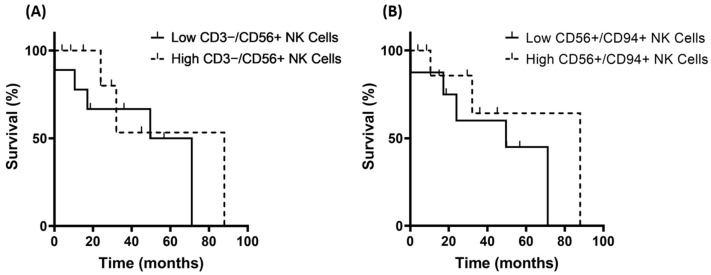
Correlation of the frequency of CD3−/CD56+ and CD56+/CD94+ NK cell subpopulations and the overall survival (OS) in patients with grade 3 glioma (low NK cells *n* = 14; high NK cells *n* = 14) as determined by the Kaplan–Meier analysis. (**A**) OS in grade 3 glioma patients with low and high frequencies of CD3−/CD56+ NK cells. (**B**) OS in grade 3 glioma patients with low and high frequencies of CD56+/CD94+ NK cells.

**Figure 9 biomedicines-11-03235-f009:**
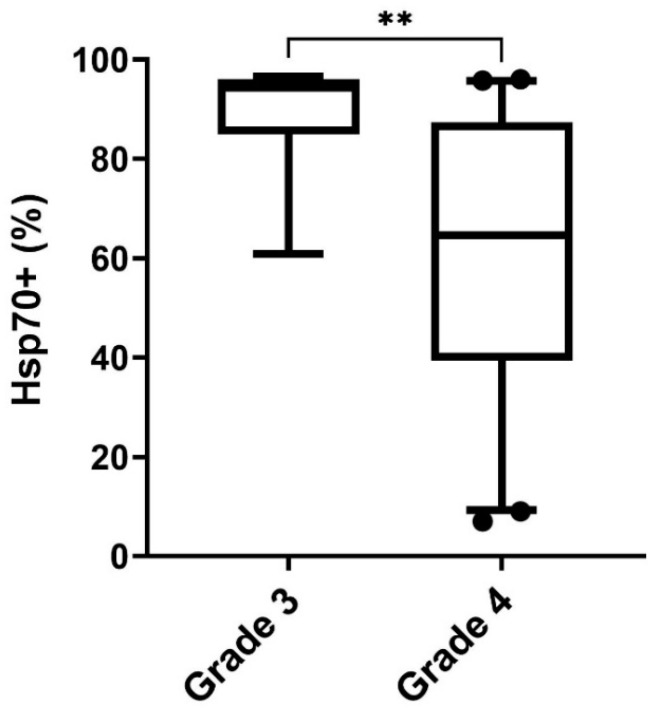
Differences between the percentages of grade 3 glioma (*n* = 6) and grade 4 GBM (*n* = 45) expressing mHsp70. Statistically significant differences ** *p* < 0.01.

**Table 1 biomedicines-11-03235-t001:** Characteristics, IDH, and 1p19q status and therapy of patients with grade 3 glioma (oligodendroglioma and astrocytoma) and grade 4 glioblastoma (GBM).

Parameters	Grade 3 Glioma		Grade 4 Glioma
Tumor histology	Oligodendroglioma	Astrocytoma	GBM
Number of patients (*n*)	15	16	99
Gender (f/m)	6/9	7/9	26/73
Age range	29–75	29–81	25–94
Median age	54	49	65
Isocitrate dehydrogenase status (IDH)	IDH mutant	IDH mutant	IDH wildtype
1p19q status	Codeleted	Non-codeleted	-
Therapy regimen	Maximal resection + RT + sequential PC(V)	Maximal resection + RT + sequential or simultaneous TMZ	Maximal resection + RT + simultaneous and sequential TMZ

Abbreviations: RT, Radiotherapy (59.4 Gy/1.8 Gy or 60 Gy/2 Gy per day, 5× per week for both grade 3 and grade 4 glioma); PCV, Procarbazine, Lomustine, Vincristine; TMZ, Temozolomide.

## Data Availability

The data presented in this study are available upon request from the corresponding author.

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
