# Peer review of "Biomarkers in Adult-Type Diffuse Gliomas: Elevated Levels of Circulating Vesicular Heat Shock Protein 70 Serve as a Biomarker in Grade 4 Glioblastoma and Increase NK Cell Frequencies in Grade 3 Glioma"

_biomedicines, 2023, doi:10.3390/biomedicines11123235_

Round 1

Reviewer 1 Report

Comments and Suggestions for Authors

HSP70 was crucial in the pathogenesis of cancer because it stabilized the production of a lot of oncogenic proteins, which in turn helped the tumor grow and survive. The authors of this study assessed the presence of Hsp70 in a group of various brain tumors. The presence of Hsp70 was examined in 130 patients' fresh tumor samples and patient serum for the study. According to the experiment results, glioblastoma and anaplastic oligodendrogliomas have much higher levels of Hsp70 in the blood than those with astrocytoma or oligodendroglioma. The techniques employed are suitable and openly disclosed. The illustrations are clear and pertinent, and the manuscript is presented in superb condition. I'm simply intrigued 

1) Have the authors had the same results in glioblastoma cell cultures versus normal astrocytes? Cultures could exclude Hsp70 activation by non-glioma-specific factors.

 2) Do the authors have evidence of a difference in expression between wild type GBM and IDH1 mutated GBM

Comments on the Quality of English Language

 Minor editing of English language required

Author Response

  • Have the authors had the same results in glioblastoma cell cultures versus normal astrocytes? Cultures could exclude Hsp70 activation by non-glioma-specific factors.

Response: This is an interesting question. Hsp70 levels in the supernatant and on the cell surface of primary astrocytes and the glioma cell line U87MG have been measured using the Hsp70 ELISA and by flow cytometry. It appeared that only cultured glioblastoma cells but not normal brain cells such as astrocytes, do express Hsp70 on the cell surface. Soluble Hsp70 could only be detected in the supernatant of cultured glioblastoma cells (U87MG) but not in the supernatant of cultured astrocytes. The data have been included as a supplementary Figure 3.

  • Do the authors have evidence of a difference in expression between wild type GBM and IDH1 mutated GBM

Response: Since all grade 4 GBM patients were IDH-wildtype this comparison has not been possible. Future studies with larger patient cohorts will study this interesting aspect.

Minor editing of English language required.

Response: The English has been corrected by the native English speaker and co-author Professor Alan Graham Pockley.

Reviewer 2 Report

Comments and Suggestions for Authors

My comments are:

1) Keyword: extracellular and membrane Hsp70. It is not clear where Hsp70 is located. It could also be located on the membrane of exosome or other vesicle.

2) It is not clear whether the figure 3A presents both free and vesicular HSP70 or only vesicular Especially because Figure 1 shows that Hsp70-exo ELISA will measure only vesicular HSP70.

3) I suggest to show individual values on the graph.

The main concern of this study is that there were no experiments performed confirming the presence of exosomes/small extracellular vesicles (e.g. western blot, electron microscopy, NTA). This is obligatory when performing experiments with exosomes.

Hsp70 is well known universal exosomal biomarker. Have the authors analyzed the relation between Hsp70 concentration and number of exosomes? How do the authors put this information into the context of their research?

Author Response

Reviewer 2

  • Keyword: extracellular and membrane Hsp70. It is not clear where Hsp70 is located. It could also be located on the membrane of exosome or other vesicle.

Response: The keywords have been corrected accordingly. The new key words are: Extracellular free and vesicular Hsp70, plasma membrane-bound Hsp70

  • It is not clear whether the figure 3A presents both free and vesicular HSP70 or only vesicular. Especially because Figure 1 shows that Hsp70-exo ELISA will measure only vesicular HSP70.

Response: This aspect has been clarified in the revised version of the manuscript. The Hsp70-exo ELISA measures both free and vesicular Hsp70 whereas the R&D systems Hsp70 ELISA measures only free Hsp70.

  • I suggest to show individual values on the graph.

Response: Individual values of the different patients cohorts have been included into the legend of Figure 3.

The main concern of this study is that there were no experiments performed confirming the presence of exosomes/small extracellular vesicles (e.g. western blot, electron microscopy, NTA). This is obligatory when performing experiments with exosomes.

Hsp70 is well known universal exosomal biomarker. Have the authors analyzed the relation between Hsp70 concentration and number of exosomes? How do the authors put this information into the context of their research?

Response: This point is well taken. Additional Figures 4C-E have been included to characterize the exosomal nature of the extracellular vesicles derived from the plasma of a patient with brain tumor regarding size, binding affinity of Hsp70 carrying exosomes to cmHsp70.1 mAb and typical exosomal surface markers such as CD9, CD81 and mHsp70.

The relation of surface bound Hsp70 on exosomes and the cmHsp70.1 mAb has been demonstrated by affinity measurements using the NanoTemper instrument. A relation of the Hsp70 concentration and the number of exosomes was not performed due to the difficulties to precisely enumerate exosomes. 

Reviewer 3 Report

Comments and Suggestions for Authors

In the manuscript by Lennartz et al., the authors describe a cohort of glioma (and healthy control) patients for whom blood measurements of Hsp70 and other circulating immune cells was performed. The authors suggest a possible link between Hsp70 levels and tumor aggressiveness, and propose it's possible use as a prognostic biomarker in glioblastoma (GBM). They also propose a possible interaction between Hsp70 levels and circulating NK cells. While there is potentially interesting data, there are several significant issues with the manuscript in its current form.

Major points:

- How diagnoses were rendered for gliomas should be more explicitly stated. How was IDH status confirmed? By IHC alone? By sequencing? Was CDKN2A/B gene status considered in IDH-mutant astrocytomas? Homozygous deletion of CDKN2A/B is now considered an independent molecular surrogate for grade 4 behavior in IDH-mutant astrocytoma. 

- Figure 1 diagrams the outline of glioma patients. It is interesting that there are no cases of grade 4 IDH-mutant astrocytoma. Were these cases specifically removed? In a cohort this size, I would expect at least some grade 4 IDH-mutant astrocytoma patients.

- Statistical analyses should be performed on all Kaplan-Meier curves and p-values indicated on all figures showing KM curves. It should also be explicitly status in the text whether the survival differences are statistically significant, which seems to only have been done for the results presented in Figure 6. It is not stated whether the differences in OS presented in Figure 7 are statistically significant or not, nor what the p-values are. 

- In many instances, it is unclear the extent to which statistical comparisons are being performed. For example, in Figure 3, it is clear that each of the glioma groups (oligodendroglioma, astrocytoma, and glioblastoma) were compared to the healthy control group. But were the glioma groups compared to each other as well? For example, were Hsp70 levels compared between oligodendroglioma and glioblastoma? This is also the case for Figure 4 and Figure 5. 

- In Figures 5, 7, and 8, the authors change their analysis and group together grade 3 oligodendroglioma and grade 3 astrocytoma as one "Grade 3" group, whereas these two tumor groups are separated in previous figures. It is unclear the rationale for doing this, which should be explained. Because oligodendroglioma and astrocytoma are disparate tumor entities with different biologic behavior and outcome (as shown in Figure 2A of this manuscript), it is inappropriate to group them together without proper justification for doing so. 

- Finally, and most importantly, in the discussion and conclusions, there are several instances of results being vastly overstated, and occasionally incorrectly stated. On lines 475-477, the authors state that "...patients with grade 4 GBM and high levels of circulating Hsp70...show a lower overall survival, compared to patients with initially low Hsp70 levels"; however, in line 385, the authors specifically state that this difference in overall survival (shown in Figure 7 without p-values) was "not statistically different". Similarly, lines 551-553 are speculative and not based on data that is shown or supported by what is presented in the manuscript. Other examples of overly strong language include: line 526, "most likely" should be replaced with "possibly"; line 529, "confirming" should be replaced with "supporting". 

Minor points:

- The introduction is very long. A more concise introduction would serve the manuscript well.

- The term "glioblastoma multiforme" is no longer recommended. Instead, just "glioblastoma" is the favored term, with the same abbreviation GBM. The word "multiforme" should be removed throughout.

- There is a large male predominance in the Grade 4 glioma group, with a male to female ratio of 73 to 26, which is higher than the reported male to female ratio of 1.6 to 1. This large male predominance should be mentioned.

- The text describes subpanels Figure 3A (line 257) and Figure 3B (line 269), but "A" and "B" are not present on Figure 3. Instead, the Figure 3 legend describes the panels as "left" and "right".

Comments on the Quality of English Language

English language quality is good with only minor editing needed.

Author Response

In the manuscript by Lennartz et al., the authors describe a cohort of glioma (and healthy control) patients for whom blood measurements of Hsp70 and other circulating immune cells was performed. The authors suggest a possible link between Hsp70 levels and tumor aggressiveness, and propose it's possible use as a prognostic biomarker in glioblastoma (GBM). They also propose a possible interaction between Hsp70 levels and circulating NK cells. While there is potentially interesting data, there are several significant issues with the manuscript in its current form.

Major points:

How diagnoses were rendered for gliomas should be more explicitly stated. How was IDH status confirmed? By IHC alone? By sequencing? Was CDKN2A/B gene status considered in IDH-mutant astrocytomas? Homozygous deletion of CDKN2A/B is now considered an independent molecular surrogate for grade 4 behavior in IDH-mutant astrocytoma. 

Response: The IDH status was determined by IHC and IDH1 and IDH2 sequencing. The technique and results have been included in the Materials and Methods and Results sections of the revised manuscript. The approach for establishing CDKN1A/B gene status was only established in 2020 in our hospital. Since the patients were recruited into the study between 2015 and 2019 this information was not available for this patient cohort.

Figure 1 diagrams the outline of glioma patients. It is interesting that there are no cases of grade 4 IDH-mutant astrocytoma. Were these cases specifically removed? In a cohort this size, I would expect at least some grade 4 IDH-mutant astrocytoma patients.

Response: In our grade 4 GBM patient cohort no grade 4 IDH-mutant astrocytoma patient was present. This is now stated in the Materials and Methods part. Patients with grade 4 IDH-mutant status were not excluded from the study.

Statistical analyses should be performed on all Kaplan-Meier curves and p-values indicated on all figures showing KM curves. It should also be explicitly status in the text whether the survival differences are statistically significant, which seems to only have been done for the results presented in Figure 6. It is not stated whether the differences in OS presented in Figure 7 are statistically significant or not, nor what the p-values are. 

Response: Statistical analysis has been performed with all Kaplan-Meier curves, but no statistical significance has been observed. Multiple comparisons were conducted among all groups shown in the study. This is now stated in the Materials and Methods section of the revised manuscript.

In many instances, it is unclear the extent to which statistical comparisons are being performed. For example, in Figure 3, it is clear that each of the glioma groups (oligodendroglioma, astrocytoma, and glioblastoma) were compared to the healthy control group. But were the glioma groups compared to each other as well? For example, were Hsp70 levels compared between oligodendroglioma and glioblastoma? This is also the case for Figure 4 and Figure 5. 

Response: Multiple statistical comparisons have been performed among all groups and statistical significant differences are indicated in the Figures.

In Figures 5, 7, and 8, the authors change their analysis and group together grade 3 oligodendroglioma and grade 3 astrocytoma as one "Grade 3" group, whereas these two tumor groups are separated in previous figures. It is unclear the rationale for doing this, which should be explained. Because oligodendroglioma and astrocytoma are disparate tumor entities with different biologic behavior and outcome (as shown in Figure 2A of this manuscript), it is inappropriate to group them together without proper justification for doing so. 

Response: A comparison of oligodendroglioma and astrocytoma grade 3 did not reveal any statistical significance. The data of this subgroup analysis have been included in supplementary Figure 2. The main goal of this study was to compare Hsp70 values and immunophenotype in grade 3 and grade 4 glioma.

Finally, and most importantly, in the discussion and conclusions, there are several instances of results being vastly overstated, and occasionally incorrectly stated. On lines 475-477, the authors state that "...patients with grade 4 GBM and high levels of circulating Hsp70...show a lower overall survival, compared to patients with initially low Hsp70 levels"; however, in line 385, the authors specifically state that this difference in overall survival (shown in Figure 7 without p-values) was "not statistically different". Similarly, lines 551-553 are speculative and not based on data that is shown or supported by what is presented in the manuscript. Other examples of overly strong language include: line 526, "most likely" should be replaced with "possibly"; line 529, "confirming" should be replaced with "supporting". 

Response: This point is also well taken. The statements in the discussion of the revised manuscript have been reviewed and changed accordingly. Overly strong statements have been deleted.

Minor points:

The introduction is very long. A more concise introduction would serve the manuscript well.

Response: The Introduction has been shortened according to the recommendation of Reviewer 3.

The term "glioblastoma multiforme" is no longer recommended. Instead, just "glioblastoma" is the favored term, with the same abbreviation GBM. The word "multiforme" should be removed throughout.

Response: This observation is well taken and the term “multiforme” has been deleted throughout the whole manuscript.

There is a large male predominance in the Grade 4 glioma group, with a male to female ratio of 73 to 26, which is higher than the reported male to female ratio of 1.6 to 1. This large male predominance should be mentioned.

Response: This has been mentioned in the description of Table 1 in the revised manuscript.

The text describes subpanels Figure 3A (line 257) and Figure 3B (line 269), but "A" and "B" are not present on Figure 3. Instead, the Figure 3 legend describes the panels as "left" and "right".

Response: This error has been corrected in the revised version of the manuscript. Elements of the figures are now described as A and B. The former Figure 3 is now Figure 4.

The authors want to thank all reviewers for their helpful suggestions.

Round 2

Reviewer 3 Report

Comments and Suggestions for Authors

The authors addressed some, but not all, of my prior critiques. 

- The authors describe that "In selected cases a positive finding was confirmed by IDH1 and IDH2 genetic sequencing" (lines 161-2). In fact, it is the cases with negative IHC that benefit from gene sequencing, not the cases with a positive IHC result. Similarly, in a series with 99 grade 4 gliomas, it is unexpected that none is IDH-mutant. This should be discussed, and particular attention should be paid to the fact that potential cases with non-cannonical IDH mutations (I.e., IDH1 R132S/C/G and or IDH2 R172 mutations might have been missed).

-It needs to be specifically stated that CDKN2A/B status was not assessed in their cohort.

- Kaplan-Meier statistics, including p-values for individual comparisons, need to be presented in the results. It is not sufficient to just describe the statistics in the materials/methods.

Comments on the Quality of English Language

As above.

Author Response

Thank you very much for taking time to review this manuscript. Please find the responses below.

Comment 1:

The authors describe that "In selected cases a positive finding was confirmed by IDH1 and IDH2 genetic sequencing" (lines 161-2). In fact, it is the cases with negative IHC that benefit from gene sequencing, not the cases with a positive IHC result. Similarly, in a series with 99 grade 4 gliomas, it is unexpected that none is IDH-mutant. This should be discussed, and particular attention should be paid to the fact that potential cases with non-cannonical IDH mutations (I.e., IDH1 R132S/C/G and or IDH2 R172 mutations might have been missed).

Answer to comment 1: This aspect has been clarified in the Materials and Methods and Results part.

Comment 2:

-It needs to be specifically stated that CDKN2A/B status was not assessed in their cohort.

Answer to comment 2: This information has been included in the Materials and Methods section.

Comment 3:

- Kaplan-Meier statistics, including p-values for individual comparisons, need to be presented in the results. It is not sufficient to just describe the statistics in the materials/methods.

Answer to comment 3: As recommended the p values have been included for the Kaplan-Meier curves in the Results part.

The English has been rerevised by a native speaker.

Round 3

Reviewer 3 Report

Comments and Suggestions for Authors

The authors have satisfactorily addressed all of my comments.

Comments on the Quality of English Language

Fine.